# Leukocyte- and Platelet-Rich Fibrin (L-PRF) Obtained from Smokers and Nonsmokers Shows a Similar Uniaxial Tensile Response In Vitro

**DOI:** 10.3390/biomedicines11123286

**Published:** 2023-12-12

**Authors:** Cesar Lara, Alejandro Bezmalinovic, Claudio García-Herrera, Susana Ríos, Loreto M. Valenzuela, Constanza E. Martínez

**Affiliations:** 1Laboratory of Biomechanics and Biomaterials, Mechanical Engineering Department, Faculty of Engineering, Universidad de Santiago de Chile, Santiago 9170022, Chile; cesar.lara.g@usach.cl (C.L.); alejandro.bezmalinovic@usach.cl (A.B.); 2School of Dentistry, Faculty of Medicine, Pontificia Universidad Católica de Chile, Santiago 8330024, Chile; susana.rios@uc.cl; 3Department of Chemical and Bioprocess Engineering, School of Engineering, Pontificia Universidad Católica de Chile, Santiago 7820436, Chile; lvalenzr@ing.puc.cl; 4Institute for Biological and Medical Engineering, Schools of Engineering, Medicine and Biological Sciences, Pontificia Universidad Católica de Chile, Santiago 7820436, Chile; 5Research Center for Nanotechnology and Advanced Materials “CIEN-UC”, Pontificia Universidad Católica de Chile, Santiago 7820436, Chile; 6Faculty of Dentistry, Universidad de los Andes, Santiago 7620086, Chile

**Keywords:** platelet-rich fibrin, smoking, mechanical tests, microscopy, electron, scanning

## Abstract

We evaluated and compared the biomechanical properties of Leukocyte-and Platelet Rich Fibrin L-PRF clots and membranes derived from smoker and nonsmoker donors. Twenty venous-blood donors (aged 18 to 50 years) were included after signing informed consent forms. L-PRF clots were analyzed and then compressed to obtain L-PRF membranes. L-PRF clot and membrane samples were tested in quasi-static uniaxial tension and the stress–stretch response was registered and characterized. Furthermore, scanning electron microscope representative images were taken to see the fibrin structure from both groups. The analysis of stress–stretch curves allowed us to evaluate the statistical significance in differences between smoker and nonsmoker groups. L-PRF membranes showed a stiffer response and higher tensile strength when compared to L-PRF clots. However, no statistically significant differences were found between samples from smokers and nonsmokers. With the limitations of our in vitro study, we can suggest that the tensile properties of L-PRF clots and membranes from the blood of smokers and nonsmokers are similar. More studies are necessary to fully characterize the effect of smoking on the biomechanical behavior of this platelet concentrate, to further encourage its use as an alternative to promote wound healing in smokers.

## 1. Introduction

For the last three decades, autologous platelet concentrates have been used in oral and medical clinical procedures to promote wound healing [1,2]. Leukocyte- and platelet-rich fibrin (L-PRF) is a biomaterial derived from an autologous blood sample; it is classified as a second generation platelet concentrate [3,4]. L-PRF is obtained from a venous blood sample without using anticoagulants, through a single centrifugation step, obtaining L-PRF clots capable of being compressed into membranes [5]. L-PRF is a polymerized fibrin network with leukocytes and activated platelets, which are critical factors for wound healing success. L-PRF membranes and clots are used in wound receptor sites to promote tissue wound healing and regeneration. Additionally, the fibrin matrix, the continuous release of cytokines, and growth factors promote important cellular biological activities. Previously, in vitro and in vivo reports have suggested that there are beneficial effects of L-PRF derived from nonsmokers in a wide range of oral clinical situations, including alveolar socket preservation, sinus lift augmentation, periodontal defect treatment, healing of chronic wounds, improving patient-related outcomes such as gingival recession treatment, and reducing postsurgical discomfort and pain [6,7,8,9,10,11,12,13,14]. In addition, L-PRF has been suggested as having a positive effect on secondary dental implant stability and leading to less bone marginal loss around implants [7,15,16].

Biomaterials used in regenerative medicine must have functional integrity to provide mechanical support for cell biological activities, improving wound healing and regeneration. At the same time, these biomaterials can release biological factors at the receptor site, acting as a temporal scaffold. The mechanical properties of L-PRF membranes have been analyzed in previous reports using uniaxial tensile [17,18,19,20,21,22] or nanoindentation tests. Other groups have compared the uniaxial tensile response of two or more L-PRF variants [18,20,21,23]. However, these studies included L-PRF derived from nonsmoker donors [24].

On the other hand, smoking has been considered to have a negative influence on wound healing and tissue regeneration due to the nicotine content and the more than 5000 toxic products generated after its combustion. It is important to remark that these compounds affect wound healing and tissue repair, negatively influencing several biological processes, including oxidative damage, changes in angiogenesis, cell proliferation, collagen synthesis, and inflammatory cell responses [25]. Consequently, previous reports have shown more recurrent postsurgical complications such as necrosis, site infection, healing delay, dehiscence, hernia, and lack of fistula and bone healing in smokers than in nonsmokers [26]. Notably, in the oral cavity, cigarette smoking represents a significant risk factor for periodontal disease and oral cancers. Nowadays, there is little scientific evidence related to smoking effects on the characteristics of L-PRF derived from smokers [27,28,29]. Moreover, the role of L-PRF obtained from smokers in stimulating wound healing during dental clinical procedures has not been studied. Therefore, this study aimed to evaluate the in vitro biomechanical properties of L-PRF, using uniaxial tensile tests and scanning electron microscopy to compare the mechanical response of L-PRF derived from smoker and nonsmoker donors.

### 1.1. Summary Box

#### 1.1.1. What Is Known?

Much in vitro evidence has demonstrated that L-PRF can increase cell proliferation, migration, angiogenesis, and differentiation in many cell types. Furthermore, it can induce anti-inflammatory activity, shifting the macrophage phenotype from M1 to M2. In vivo, growing evidence has suggested that L-PRF improves periodontal tissue healing and regeneration. This evidence involves L-PRF derived from nonsmokers, and only a few in vitro studies have explored the effects of smoking on the properties of L-PRF.

#### 1.1.2. What This Study Adds

This research aimed to evaluate and compare the structure and tensile properties of L-PRF derived from smokers/nonsmokers. The results suggest that L-PRF membranes have higher tensile strength than L-PRF clots and similar structural features. However, differences were not observed between smokers/nonsmokers.

## 2. Materials and Methods

### 2.1. Isolation of L-PRF Membranes and Clots

Venous blood samples were obtained from twenty healthy donors aged 18 to 50 years old. Ten donors per group were included in two groups: nonsmokers and smokers. People who smoked at least ten cigarettes per day were included. Cotinine was used to verify smoking habits in L-PRF exudates, as reported previously [27]. All procedures applied the guidelines set by the Health Sciences Ethics Committee and all participants signed informed consent forms (ethical approval ID17076007). From each donor, 18 ml of blood was collected using two silica-coated sterile tubes (red cap BD Vacutainer blood collection tubes, Beckton Dickinson, Franklin Lakes, NJ, USA, 9 mL per tube) without anticoagulants. Tubes were immediately centrifuged at 408 xG (2700 rpm = 700 RCF) for twelve minutes at room temperature using a laboratory centrifuge (EBA-200, Hettich, Kirchlengern, Germany). After centrifugation, each L-PRF clot was isolated from the middle of the tube using sterile forceps. One clot per donor was gently compressed by gravity using a sterile surgical box and a 225 g steel plate (IntraSpin, Biohorizons, Birmingham, AL, USA) for five minutes to obtain L-PRF membranes of approximately 1 mm thickness (Figure 1A–F).

At least one L-PRF clot and one L-PRF membrane were successfully obtained from each donor and tested in uniaxial tension to group-wise evaluate the uniaxial tensile mechanical properties of L-PRF. Membranes and clots were preserved in sterile tubes containing a DMEM cell culture medium at 4 °C until their testing.

### 2.2. Scanning Electron Microscopy (SEM)

For SEM analysis, six L-PRF membranes (three derived from smokers and three from nonsmokers) and six L-PRF clots (three derived from smokers and three from nonsmokers) were included. L-PRF membranes and clots were fixed with 2.5% glutaraldehyde and postfixed with 1% osmium tetroxide. Then, two sections were derived from each sample, one including the buffy coat holding the highest cell concentration and the other section distal to the buffy coat (Figure 1). Later, samples were dehydrated in ethanol solutions, then critical point dried (NEWMED FJVA critical drying system model CPDS-20202) and gold metalized (VARIAN VACUUM Evaporator PS10E, Varian Inc., Palo Alto, CA, USA). Finally, representative images of each sample at two different magnifications were registered using a Hitachi TM3000 (Hitachi, high-technologies corporation, Chiyoda, Tokyo, Japan) microscope at 15 kV.

### 2.3. Preparation of L-PRF Samples for Uniaxial Tensile Testing

Fourteen L-PRF membranes and fourteen L-PRF clots (seven derived from smokers and seven from nonsmokers) were analyzed. From each L-PRF clot and membrane isolated, a rectangular sample of 10 to 15 mm in length and 5 to 6.5 mm in width (w_0_) was cut using a scalpel (Figure 2A,B). The sample thickness (h_0_) was measured as the simple average of three recordings (at relative distances of 25%, 50%, and 75% of the sample length) using a Keyence (Keyence, Minato City, Japan) model LS-7070M high-speed optical digital micrometer (accuracy, 50 μm).

### 2.4. Uniaxial Tensile Tests

A mono-column testing machine (Instron 3342, Instron, Norwood, MA, USA) was used to perform uniaxial tension tests on L-PRF samples. Briefly, a monotonically increasing displacement ∆L was applied to the crosshead of the device until the sample rupture. Specimens were fixed in place using stainless-steel grips aided by applying a cyanoacrylate-based adhesive and mechanical compression and were immersed in phosphate-buffered saline (PBS 1x) at a physiological temperature of 38 ± 1 °C during the tests (Figure 2C). The instantaneous axial force (F) was measured with a 10N-capacity load cell, and the displacement was applied at a rate of 2 mm/min to attain quasi-static conditions.

Each force–displacement (F-∆L) response curve obtained was turned into its Cauchy (or true) stress-versus-stretch (σ-λ) counterpart, with Cauchy stress defined as σ = F λ/(w_0_h_0_), and stretch given by λ = 1 + (∆L/L_0_), where L_0_, w_0_, and h_0_ are the sample initial (or undeformed) in-between-grips length and width (Figure 2D).

From the uniaxial stress–stretch (σ-λ) curves, three mechanical parameters were determined, which allowed for the quantification of the material stiffening at low and high levels of deformation: the slope (elastic modulus) at low-strain E_1_ (from the first linear stress–stretch zone of the curve, where 1 ≤ λ ≤ λ_1_) and the elastic modulus at high-strain E_2_ (slope of a second linear zone where λ_2_ ≤ λ < λ_r_). Likewise, the point (λ_r_, σ_r_) shows the stretch–stress pair for which the sample reaches rupture, corresponding to the stretch-at-failure and tensile-strength, respectively.

### 2.5. Statistical Analysis

The values of mechanical parameters derived from stress–stretch curves were expressed as mean ± SD (standard deviation). Data normality was confirmed using the Shapiro–Wilk test. *T*-tests or ANOVA tests were used to establish the existence of statistically significant differences between the mean of the smoker and nonsmoker study groups, using GraphPad Prism 8.0 software (GraphPad Software Inc., San Diego, CA, USA), considering a two-tailed probability (*p*-value) of finding a set of data as (or more) extreme that complies with the null hypothesis of *p* ≤ 0.05 as statistically significant.

## 3. Results

### 3.1. Morphological Characterization

Scanning electron microscope representative images of L-PRF clots (Figure 3) and L-PRF membranes (Figure 4) derived from smokers and nonsmokers were analyzed. Two areas were seen in both preparations: the buffy coat, where a high cell concentration is found, and another distal to the buffy coat.

A similar cell content was seen in both L-PRF clots recovered from nonsmokers and smokers. Red blood cells, leukocytes, and platelets were identified as shown (Figure 3A,B,D,E). In both groups, inactive and active platelets were seen (Figure 3G,H,J,K). A higher cell concentration was observed in the buffy coat zone compared to the distal zone of the buffy coat (Figure 3A,B,D,E,G,H,J,K). A considerable number of fibrin networks were seen in both groups at the distal zone to the buffy coat, showing high-density fibrin fibers with similar characteristics in both nonsmokers and smokers (Figure 3C,F,L,I).

Regarding L-PRF membranes, a similar cell content was seen in samples recovered from nonsmokers and smokers. Similarly to the L-PRF clots, a higher cell concentration was evidenced in the buffy coat zone, where red blood cells, leukocytes, and platelets were identified as shown (Figure 4A,D,G,J). Distally to the buffy coat, fibrin networks which were highly ordered and more packed compared to L-PRF clots were seen in both groups with remarkably similar characteristics (Figure 4B,C,E,F,H,I,K,L).

### 3.2. Uniaxial Stress–Stretch Curves and Mechanical Parameters

The average (undeformed) geometrical properties of L-PRF samples were similar, since differences between L-PRF membranes or L-PRF clots derived from nonsmokers/smokers were statistically insignificant (Figure 5). As expected, L-PRF membranes had a lower thickness than L-PRF clots due to the compression during processing (Figure 5).

Statistical differences between smokers and nonsmokers, among either L-PRF membranes or clots, were not seen in all mechanical parameters evaluated (Figure 6). However, L-PRF membranes from both smokers/nonsmokers showed a (significantly) higher stiffness than L-PRF clots (Figure 6B–D).

The elastic modulus at a low strain (E1) of L-PRF membranes derived from nonsmokers (23.9 +/− 19.97 kPa) was higher than L-PRF clots from nonsmokers/smokers (3.7 ± 3.1 kPa/4.4 ± 3.5 kPa; * *p* = 0.012) (Figure 6B). Likewise, L-PRF membranes derived from nonsmokers/smokers had a higher elastic modulus at a high strain (E2) (187.6 ± 82.73 kPa/181.8 ± 74.48 kPa) compared to L-PRF clots derived from nonsmokers/smokers (30.2 ± 16.7 kPa/37.8 ± 17.6 kPa; *** *p* = 0.0003) (Figure 6C). 

The transition stretch (λc) of nonsmoker/smoker L-PRF samples was similar (2.0 ± 0.3/2.0 ± 0.3 for membranes and 2.2 ± 0.4/2.2 ± 0.3 for clots) as well as the stretch-at-failure (λr) (4.3 ± 0.4/4.2 ± 0.4 for membranes and 4.3 ± 0.4/4.3 ± 0.3 for clots).

Finally, tensile strength (σr) was significantly higher in L-PRF membranes from nonsmokers/smokers (675.8 ± 409.9 kPa/621.1 ± 466.5 kPa) than in L-PRF clots from nonsmokers/smokers (86.7 ± 73.5 kPa/109.9 ± 71.14 kPa; * *p* = 0.013, ** *p* = 0.009) (Figure 6D).

## 4. Discussion

L-PRF has been considered the second generation of platelet concentrates with proven beneficial effects in dental clinical procedures for enhancing wound healing and regeneration in nonsmokers. In this study, L-PRF mechanical properties and morphological features derived from smokers (who smoke at least ten cigarettes per day) were analyzed and compared to nonsmokers, showing comparable properties between groups. Our results show a similar trend to those previously reported by Srirangarajan et al. [28] reporting higher uniaxial tensile strength in L-PRF membranes derived from nonsmokers (157 ± 0.07 kPa) compared to smokers (112 ± 0.07 kPa), but without statistical differences. To our knowledge, Srirangarajan’s study is the only previous report on the tensile properties of L-PRF membranes derived from smokers.

In the same way, Madurantakam et al. [17] performed uniaxial tensile tests on L-PRF membranes, reporting higher elastic modulus values (470 ± 107 kPa) and a lower stretch-at-failure (2.17 ± 0.39 KPa) than our results, probably due to stiffening and viscous effects caused by a higher load-rate (10.0 mm/min) in the experimental setting. On the other hand, [19] Ockerman et al. evaluated the effect of using anticoagulants on the mechanical properties of L-PRF membranes derived from nonsmokers, evidencing a lower elastic modulus (70 ± 1 kPa), tensile strength (290 ± 80 kPa) and stretch-at-break (2.8 ± 0.4) compared to our results. Ravi et al. [23] reported relatively high values of tensile strength (290.076  ±  5.68 MPa) and modulus of elasticity (98.01  ±  7.43 MPa) in L-PRF membranes derived from nonsmokers. However, a non-standard L-PRF-obtaining protocol was used (1960 rpm or 708 xG). In the same way, Lee et al. [22] reported uniaxial “tensile strength” for L-PRF membranes from nonsmokers in units of force (1.7 ± 0.7 N). Since our L-PRF membranes from nonsmokers have an initial cross-sectional area of approximately 4.4 mm^2^, their tensile strength is equivalent to a load-at-failure of 0.7 ± 0.2 N. More recently, Simões-Pedro et al. [20] contrasted the uniaxial tensile response of three L-PRF variants, describing for L-PRF membranes a “maximum traction” of 1 kPa and an “average traction” of 0.7 kPa, similar to our results. Pascoal et al. [21] studied the uniaxial tensile response of L-PRF membranes from nonsmokers, reporting a mean “average traction” of 19.2 kPa, like the slope at low strain evidenced in our study, and a mean “maximal traction” of 42.5 kPa, in agreement with the tensile strength of L-PRF published by Khorshidi et al. [18].

Most previous reports evaluating the mechanical properties of L-PRF membranes were derived only from nonsmokers, and differences in the testing protocol could explain differences between our results and those previously discussed. For instance, samples were analyzed at room temperature and were not submerged in PBS during the uniaxial tensile tests. In contrast, in our study, L-PRF samples were kept in a (thermally regulated) humid environment to simulate the physiological conditions of this biomaterial.

For the first time, the mechanical properties of L-PRF clots derived from nonsmokers/smokers and L-PRF membranes from smokers are being reported. L-PRF clots from nonsmokers/smokers evidenced lower stiffness and tensile strength compared to L-PRF membranes. These results support the increasing use and preference for L-PRF membranes in the treatment of periodontal defects, sinus elevation, ridge preservation, application before implant placement, and adjunctive therapy in gingival recession treatments instead of L-PRF clots [15,16,30,31,32,33,34].

Furthermore, our in vitro results suggest that L-PRF membranes and clots derived from smokers do not significantly change in cellular content and fibrin net disposition, either in the stiffness at low and high strain (elastic-moduli) or the stress and elongation-at-break, compared to nonsmokers.

The limitations of this study included the sample size determination due to the need for previous in vitro reports, including L-PRF membranes/clots from smokers. However, similar studies in nonsmokers have included a similar sample size. Donors who smoke at least ten cigarettes per day were included. However, the mechanical properties of L-PRF derived from heavy smokers (19 cigarettes or more per day) [35] were not evaluated. Future studies could include donors with different levels of smoking.

A few previous reports have suggested contradictory results regarding the effects of smoking on fibrin clot structure. In 2010, Barua et al. reported changes in fibrin architecture evaluated by SEM in venous blood clot samples, including eight nonsmoker donors and twelve healthy smokers after overnight smoking abstinence and immediately after smoking two cigarettes. They observed significantly thinner fibers and more strands/µm^2^ than fibrin clots from pre-smoking and nonsmoking donors [36]. Similarly, Stępień et al. reported an increase in fibrin fiber branching after smoking cessation in blood clots derived from current smoker donors. However, they did not find changes in the fibrin fiber diameter [37]. Nonetheless, these studies collected blood in the presence of anticoagulants to obtain the blood plasma and induce in vitro clotting using thrombin. As reported before, these biochemical modifications could alter the fibrin structure [36,37]. Our L-PRF-obtaining protocol does not include the use of biochemical modifiers of the coagulation cascade, and clotting occurs physiologically [38,39]. The use of anticoagulated plasma and other experimental design conditions could explain the differences in SEM observations between this study and previous reports on the clot structure of smokers.

Finally, a recent in vitro report from our research group suggested that smoking does not affect biological activities triggered by conditioned media recovered from L-PRF. We demonstrated that an L-PRF-conditioned medium derived from membranes/clots from smokers stimulates periodontal ligament mesenchymal stromal cell proliferation and migration, similar to L-PRF-conditioned media derived from nonsmokers [27]. We even observed a recovery of the harmful effect of nicotine treatment in cell proliferation/migration in the presence of an L-PRF-conditioned medium obtained from smokers [27]. The results of this in vitro study, added to our previous in vitro report, suggest that L-PRF could be a beneficial autologous tool to stimulate wound healing in smokers and avoid postsurgical complications, as previously reported. However, in vivo studies are needed to establish a positive effect of L-PRF membranes in dental clinical procedures such as alveolar ridge preservation, sinus lift, periodontal defects, and gingival recession treatments.

It is important to emphasize that cell biological activities are affected by smoking, causing poor healing; for that reason, smokers may be more likely than nonsmokers to develop complications after surgical procedures, such as infection, alveolitis post-extraction, wound and flap necrosis, dehiscence, and a decrease in wound tensile strength. L-PRF use in surgical procedures in smokers could be beneficial due to its similar in vitro characteristics in nonsmokers. However, more studies are necessary to prove this beneficial effect.

## 5. Conclusions

Our results suggest that smoking does not affect the in vitro biomechanics of L-PRF membranes and L-PRF clots, evidenced by a similar fibrin structure and cells and comparable uniaxial tensile responses. However, further studies are required to encourage the use of L-PRF as an alternative to promote wound healing in smokers.

## Figures and Tables

**Figure 1 biomedicines-11-03286-f001:**
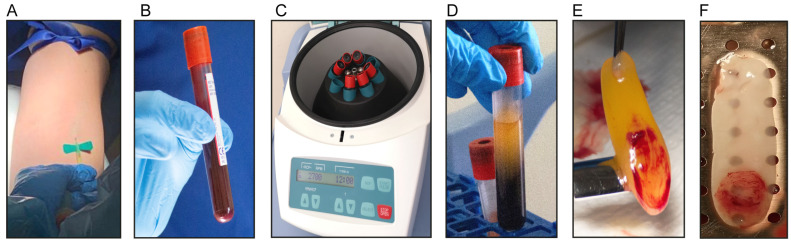
Procedure for isolation of L-PRF membranes and clots. (**A**,**B**) Blood harvesting; (**C**) tube centrifugation; (**D**,**E**) L-PRF clot isolation; (**F**) L-PRF membrane.

**Figure 2 biomedicines-11-03286-f002:**
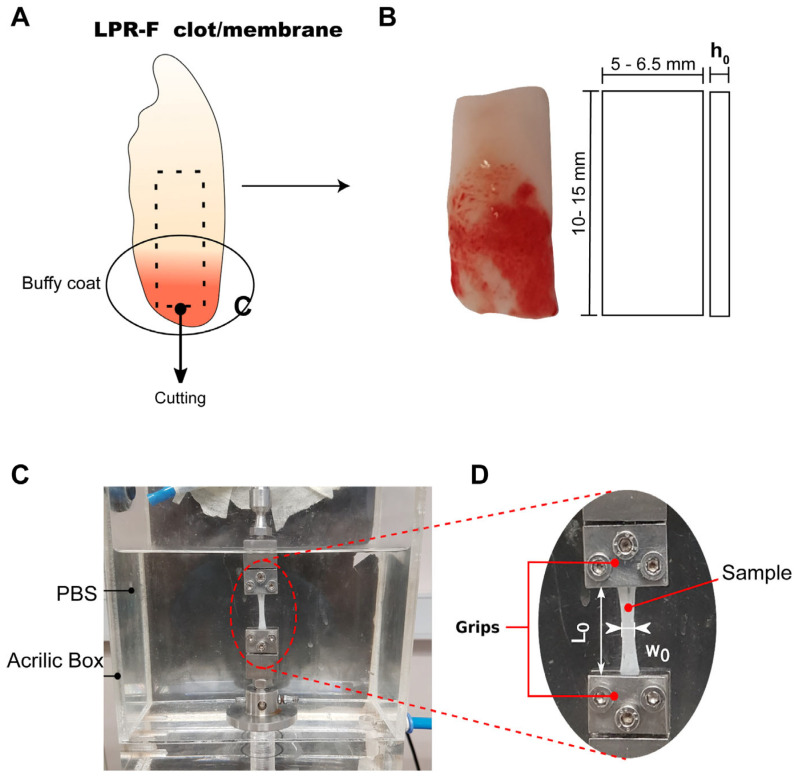
L-PRF sample preparation for uniaxial tensile testing. (**A**) Cutting of L-PRF clot/membrane; (**B**) L-PRF clot/membrane after cutting; (**C**) uniaxial tensile test setup; (**D**) initial geometry of L-PRF samples.

**Figure 3 biomedicines-11-03286-f003:**
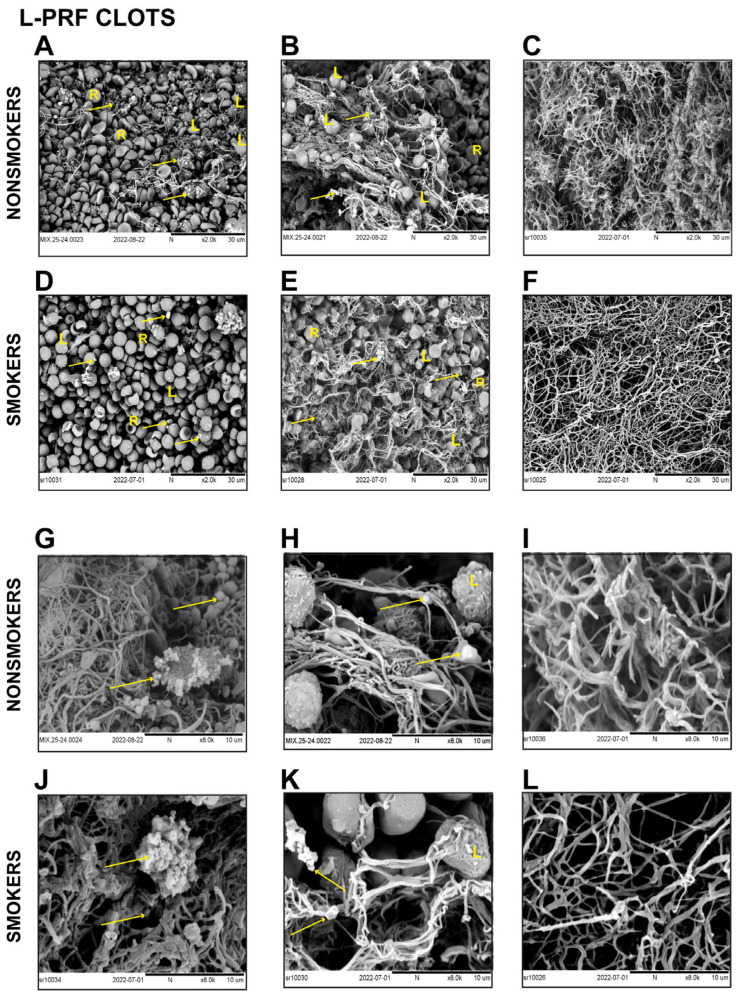
Representative SEM images of L-PRF clots derived from nonsmokers and smokers showed similar cell and fibrin characteristics. L-PRF clots from nonsmokers at the buffy coat (**A**,**B**) and at the distal zone to the buffy coat (**C**). Images at higher magnification in the buffy coat (**G**,**H**) and at the distal site (**I**). L-PRF clots from smokers at the buffy coat (**D**,**E**) and the distal zone (**F**). Images at higher magnification at the buffy coat (**J**,**K**) and the distal area (**L**). R = red blood cell, L = leucocyte. Yellow arrows indicate platelets.

**Figure 4 biomedicines-11-03286-f004:**
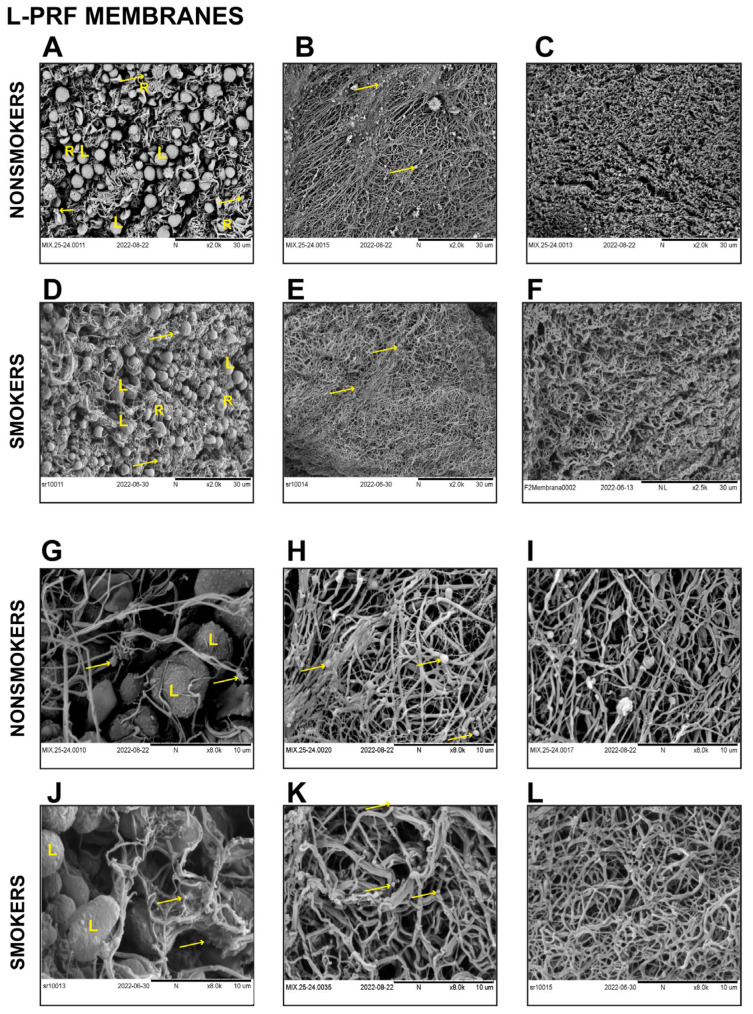
Representative SEM images of L-PRF membranes derived from nonsmokers and smokers show similar cell and fibrin characteristics. L-PRF membranes from nonsmokers at the buffy coat (**A**) and the distal zone to the buffy coat (**B**,**C**). Images at higher magnification in the buffy coat (**G**) and the distal area (**H**,**I**). L-PRF membranes from smokers at the buffy coat (**D**) and the distal zone (**E**,**F**). Images at higher magnification at the buffy coat (**J**) and the distal area (**K**,**L**). R = red blood cell, L = leucocyte. Yellow arrows indicate platelets.

**Figure 5 biomedicines-11-03286-f005:**
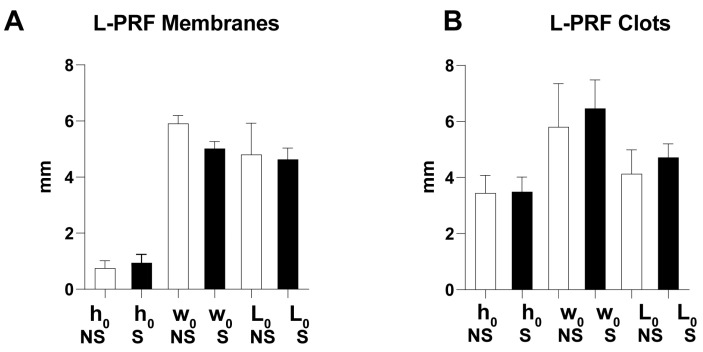
L-PRF membranes and L-PRF clot samples derived from smokers/nonsmokers evidenced similar geometrical properties. (**A**) L-PRF membranes; (**B**) L-PRF clots. Thickness (h_0_), width (w_0_), and length (L_0_). NS = nonsmokers, S = smokers.

**Figure 6 biomedicines-11-03286-f006:**
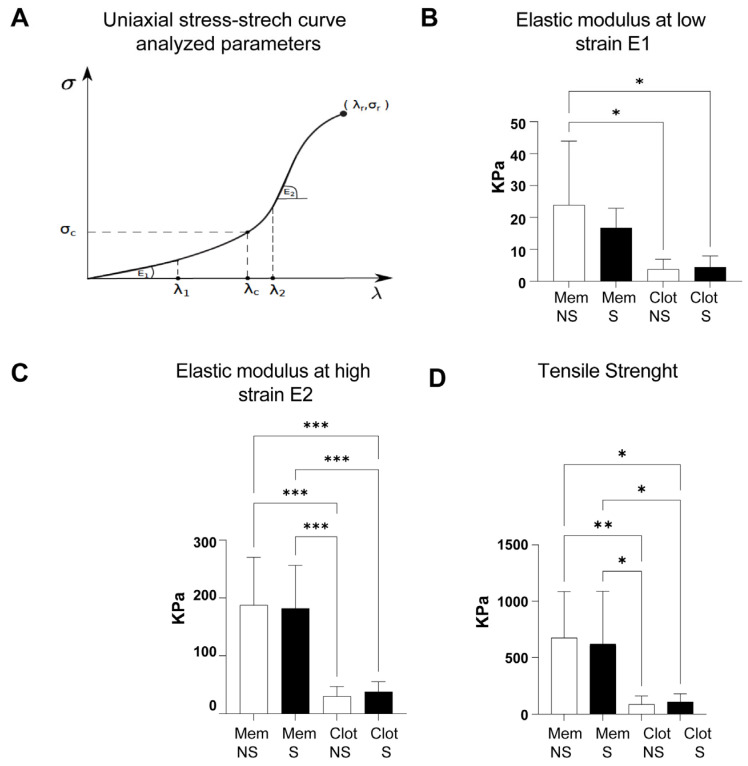
In uniaxial stress–stretch curves, L-PRF membranes and L-PRF clots derived from nonsmokers/smokers exhibit similar mechanical parameters. (**A**) Mechanical parameters defined on a typical uniaxial stress–stretch (σ-λ) curve: elastic modulus at low strains (E1), elastic modulus at high strains (E2), transition point (λc, σc) and rupture point (λr, σr). (**B**–**D**) Mechanical parameters (in kPa) of L-PRF membranes (Mem) and clots derived from smokers/nonsmokers: (**B**) elastic modulus at a low strain E1 (* *p* = 0.012); (**C**) elastic modulus at a high strain E2 (*** *p* = 0.0003); (**D**) tensile strength (* *p* = 0.013, ** *p* = 0.009). NS = nonsmokers, S = smokers.

## Data Availability

The data presented in this study are available on request from the corresponding authors.

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
