# Peer review of "Leukocyte- and Platelet-Rich Fibrin (L-PRF) Obtained from Smokers and Nonsmokers Shows a Similar Uniaxial Tensile Response In Vitro"

_biomedicines, 2023, doi:10.3390/biomedicines11123286_

Round 1
Reviewer 1 Report
Comments and Suggestions for Authors
Nice concise study. I would recommend expanding on the rationale why the authors believed the mechanical properties would differ between the 2 groups. This discussion would add value to the manuscript.
Author Response
Thank you for allowing us to submit a revised draft of our manuscript. Thank you very much for taking the time to review this manuscript. We are grateful to the reviewers for their insightful comments on this paper. We have been able to incorporate changes to reflect most of the suggestions provided by the reviewers. We have highlighted the changes within the manuscript version 2.
Reviewer 1:
Nice, concise study. I would recommend expanding on the rationale why the authors believed the mechanical properties would differ between the 2 groups. This discussion would add value to the manuscript.
A/ Thank you for pointing this out. We agree and added the following paragraph to the discussion:
Few previous reports have suggested contradictory results of the smoking effects on the Fibrin clot structure. In 2010, Barua et al. reported changes in fibrin architecture evaluated by SEM in venous blood clot samples, including eight nonsmokers donors and twelve healthy smokers after overnight smoking abstinence and immediately after smoking two cigarettes. They observed significantly thinner fibbers and more strands/ µm2 than fibrin clots from pre-smoking and nonsmoking donors [41]. Similarly, StÄ™pieÅ„ et al. reported an increase of fibrin fiber branching after smoking cessation in blood clots derived from current smoker donors. However, they did not find changes in the fibrin fiber diameter [42]. Nonetheless, these studies collected blood in the presence of anticoagulants to get the blood plasma and induce in vitro clotting using thrombin. As reported before, these biochemical modifications could alter the fibrin structure [41,42]. L-PRF obtaining protocol does not include the use of biochemical modifiers of the coagulation cascade, and clotting occurs physiologically. The use of anticoagulated plasma and other experimental design conditions could explain the differences in SEM observations between this study and previous reports on smoker’s clot structure.

Reviewer 2 Report
Comments and Suggestions for Authors
In the original article entitled „Leukocyte and platelet-rich fibrin (L-PRF) obtained from smokers and nonsmokers shows a similar uniaxial-tensile response in vitro” by Dr Lara et al., the Authors investigated biomechanical properties of L-PRF clots and membranes derived from healthy blood donors including smokers and nonsmokers. The Authors did not find any differences between smokers and nonsmokers. In my opinion, the study is interesting and I am fully convinced that presenting negative results is very important for increasing a knowledge in the scientific community. On the other hand, I have several comments listed below.
1. The sample size is small (n=10 for each subgroup), which was admitted by the Authors in the Limitations of the Study. However, for scanning microscope studies the subgroup size was even smaller (n=3 for each subgroup). What was a rationale for a selection of specific donors out of the both the subgroups? Was it randomised?
2. I have doubts regarding statistical analysis. “T-test or ANOVA tests were used to establish the existence of statistically significant differences between the median of the smoker and nonsmoker study groups”. Both the tests are parametric tests and as such, test rather differences between the means not the medians. I do not think that the SEM should be used as a measure of dispersion. We have various individuals (donors) not just repetitions coming from the same biological material, therefore SD (standard deviation) should be presented instead.
3. Please give number of donors or number of experiments in the caption of figures 5 and 6.
4. Please delete “Figure 3” and “Figure 4” in the bottom of respective figures.
5. Please assure that there is a space between a text and the reference number in the square bracket. There is a lack of spaces in some places in the text e.g. lines 76, 319, 327. Double dot is placed in the line 61-62.
Author Response
Thank you for allowing us to submit a revised draft of our manuscript. Thank you very much for taking the time to review this manuscript. We are grateful to the reviewers for their insightful comments on this paper. We have been able to incorporate changes to reflect most of the suggestions provided by the reviewers. We have highlighted the changes within the manuscript version 2.
Reviewer 2:
In the original article entitled „Leukocyte and platelet-rich fibrin (L-PRF) obtained from smokers and nonsmokers shows a similar uniaxial-tensile response in vitro” by Dr Lara et al., the Authors investigated biomechanical properties of L-PRF clots and membranes derived from healthy blood donors including smokers and nonsmokers. The Authors did not find any differences between smokers and nonsmokers. In my opinion, the study is interesting, and I am fully convinced that presenting negative results is very important for increasing a knowledge in the scientific community. On the other hand, I have several comments listed below.
- The sample size is small (n=10 for each subgroup), which was admitted by the Authors in the Limitations of the Study. However, for scanning microscope studies the subgroup size was even smaller (n=3 for each subgroup). What was a rationale for a selection of specific donors out of the both the subgroups? Was it randomised?
A/ Thanks a lot for your consideration. We agree that the sample size was small in this study. Based on scanning electron microscope experiments, we analyzed only three samples per group, considering the buffy coat and the distal zone of the buffy coat. Samples were not randomized and were identified according to the group to be analyzed. However, samples were identified with a random number before and during sample preparation and during visualization in the microscope to avoid bias in the descriptive analyses of each image. SEM analysis was only illustrative, and three samples were included according to the following previous reports:
Miron, R.J., Xu, H., Chai, J. et al. Comparison of platelet-rich fibrin (PRF) produced using three commercially available centrifuges at both high (~ 700 g) and low (~ 200 g) relative centrifugation forces. Clin Oral Invest 24, 1171–1182 (2020). https://doi.org/10.1007/s00784-019-02981-2
Simões-Pedro, M.; Tróia, P.M.B.P.S.; dos Santos, N.B.M.; Completo, A.M.G.; Castilho, R.M.; de Oliveira Fernandes, G.V. Tensile Strength Essay Comparing Three Different Platelet-Rich Fibrin Membranes (L-PRF, A-PRF, and A-PRF+): A Mechanical and Structural In Vitro Evaluation. Polymers 2022, 14, 1392. https://doi.org/10.3390/polym14071392
Oliveira, M.N., Varela, H.A., Nascimento, R.M. et al. Injectable Platelet-Rich Fibrin in Contact with Bone Substitutes, Porous Zirconia, or Laser-Textured Implant Surfaces: A Detailed Morphological Analysis. Biomedical Materials & Devices (2023). https://doi.org/10.1007/s44174-023-00094-9
I have doubts regarding statistical analysis. “T-test or ANOVA tests were used to establish the existence of statistically significant differences between the median of the smoker and nonsmoker study groups.” Both the tests are parametric and as such, test differences between the means not the medians. I do not think that the SEM should be used as a measure of dispersion. We have various individuals (donors) not just repetitions coming from the same biological material, therefore SD (standard deviation) should be presented instead.
Thanks to the reviewer for pointing out this inconsistent issue. We apologize for making a typing mistake using the word median in the paragraph. We used the mean for the analysis and clarified and corrected it in the new version of the manuscript. On the other hand, we agree with you about choosing SD against SEM. All graphs and results sections were updated, including SD against SEM. Thanks a lot for your suggestion; this is a significant improvement in the manuscript. Changes were highlighted in the manuscript.
Please give number of donors or number of experiments in the caption of figures 5 and 6
Thanks for your question. In figures 5 and 6, 14 donors were included, 7 for the smoker and 7 for the nonsmoker groups. Three measurements and recordings per sample were included in the data analysis.
- Please delete “Figure 3” and “Figure 4” in the bottom of respective figures.
Thanks, the word “Figure” in each figure was deleted.
- Please assure that there is a space between a text and the reference number in the square bracket. There is a lack of spaces in some places in the text e.g. lines 76, 319, 327. Double dot is placed in the line 61-62
Thanks a lot, spaces between the reference number and square bracket were revised and double dots were eliminated. These changes were highlighted in the manuscript.
Round 2
Reviewer 2 Report
Comments and Suggestions for Authors
The Authors took into consideration all my comments and suggestions. In my opinion, the paper has been improved. I have no further comments.